# Prevention of Initial Bacterial Attachment by Osteopontin and Other Bioactive Milk Proteins

**DOI:** 10.3390/biomedicines10081922

**Published:** 2022-08-09

**Authors:** Mathilde Frost Kristensen, Esben Skipper Sørensen, Yumi Chokyu Del Rey, Sebastian Schlafer

**Affiliations:** 1Department of Dentistry and Oral Health, Section for Oral Ecology and Caries Control, Aarhus University, 8000 Aarhus, Denmark; 2Department of Molecular Biology and Genetics, Aarhus University, 8000 Aarhus, Denmark

**Keywords:** actinomyces, bacterial adhesion, biofilms, caseins, dental caries, lactobacillus, microfluidic device, osteopontin, streptococcus

## Abstract

A considerable body of work has studied the involvement of osteopontin (OPN) in human physiology and pathology, but comparably little is known about the interaction of OPN with prokaryotic cells. Recently, bovine milk OPN has been proposed as a therapeutic agent to prevent the build-up of dental biofilms, which are responsible for the development of caries lesions. Bioactive milk proteins are among the most exciting resources for caries control, as they hamper bacterial attachment to teeth without affecting microbial homeostasis in the mouth. The present work investigated the ability of OPN to prevent the adhesion of three dental biofilm-forming bacteria to saliva-coated surfaces under shear-controlled flow conditions in comparison with the major milk proteins α-lactalbumin, β-lactoglobulin, αs1-casein, β-casein and κ-casein, as well as crude milk protein. OPN was the most effective single protein to reduce the adhesion of *Actinomyces naeslundii*, *Lactobacillus paracasei subsp. paracasei* and *Streptococcus mitis*. β-casein and crude milk protein also had a pronounced effect on all three species, which suggests binding to different microbial surface structures rather than the blocking of a specific bacterial adhesin. Bioactive milk proteins show potential to delay harmful biofilm formation on teeth and hence the onset of biofilm-related oral disease.

## 1. Introduction

Osteopontin (OPN) is a highly phosphorylated protein present in most mammalian tissues and body fluids, with the highest concentration found in milk [1]. A considerable body of work has explored the multifaceted involvement of OPN in a variety of biological processes, such as cell adhesion, migration and survival, bone remodeling, inflammation and wound-healing [2]. Much less is known about the interaction of OPN with microbial cells. The protein binds avidly to a range of different bacteria, and it has been shown to act as an opsonin that promotes macrophage phagocytosis during bone repair [3]. In the oral cavity, the high affinity of OPN for bacterial cells may be exploited for therapeutic purposes. The administration of OPN isolated from bovine milk has been shown to hamper bacterial adhesion, which delays biofilm formation on teeth and may reduce the occurrence of dental caries or periodontal disease [4,5,6].

Dental caries and periodontitis are amongst the most prevalent diseases of humankind, with significant economic and health impacts worldwide [7]. Both conditions can be prevented by minimizing biofilm accumulation on tooth surfaces, which is currently achieved by mechanical cleaning and antimicrobial adjuncts [8]. Antimicrobial agents seek to kill pathogenic bacteria in dental biofilms, but they also affect commensal microorganisms on soft tissues that contribute to microbial homeostasis. The indiscriminate killing of these commensal bacteria may lead to a disequilibrium in the oral microbiota and a higher risk of developing systemic diseases, such as diabetes and hypertension [9]. Research is therefore focusing on novel strategies for biofilm control that aim at interfering with the mechanisms of biofilm formation instead of simply eradicating oral microorganisms.

Milk products have repeatedly been reported to have caries-preventive properties, and part of this effect may be due to the action of milk proteins [10]. Other than OPN, α-lactalbumin, β-lactoglobulin and several proteins from the casein family have been shown to reduce bacterial attachment in different studies [11,12,13,14,15,16]. However, in most investigations, bacterial adhesion was tested during static incubation with a saliva-free inoculation medium, and hence under conditions that did not accurately mimic the situation in the mouth, where bacterial attachment takes place under constant saliva flow.

The aim of the present work was therefore to investigate the effects of OPN and the principal milk proteins α-lactalbumin, β-lactoglobulin, α_s1_-casein, β-casein and κ-casein, as well as crude milk protein, on bacterial adhesion in a shear-controlled microfluidic device [17] providing a salivary flow rate representative of the oral cavity [18]. To cover a range of different dental biofilm species, the primary colonizers *Streptococcus mitis* and *Actinomyces naeslundii*, as well as the cariogenic *Lactobacillus paracasei subsp. paracasei*, were chosen for the experiments. The null hypothesis was that none of the investigated proteins would reduce bacterial adhesion of any of the three tested strains compared to control treatment with phosphate-buffered saline (PBS).

## 2. Materials and Methods

### 2.1. Purification of Milk Proteins

Skim milk protein was prepared by first removing the milk fat from fresh bovine milk by centrifugation at 4200× *g*. Thereafter the skimmed milk was dialysed against Millipore water over night in a dialysis tube with cut-off of 12 kDa (Medicel Mebranes Ltd., London, UK) to remove lactose and other small non-protein material, followed by freeze-drying. The caseins αs1-, β- and κ-casein were purified from fresh bovine milk by size exclusion and cation exchange chromatography, as described previously [19,20]. α-lactalbumin and β-lactoglobulin were purified from a whey protein isolate (Arla Foods Ingredients Group P/S, Viby J, Denmark) by reverse-phase HPLC on a Vydac C18 column (The Separations Group, Hesperia, CA, USA). The protein was separated in 0.1% trifluoroacetic acid (TFA; Sigma Aldrich, Søborg, Denmark) (Buffer A) and eluted with a gradient of 60% acetonitrile (Sigma Aldrich) in 0.1% TFA (buffer B) (Gradient: 0–5 min, 0% B; 5–65 min, 98%B; 65–70 min, 98% B; 70–72 min, 0% B). The column was operated at a flowrate of 0.85 mL/min at 40 °C and the proteins were monitored in the effluent by measuring the absorbance at 226 nm. The identity and purity of the proteins was determined by Tris-tricine gel electrophoresis and the fractions of interest were pooled and lyophilized. OPN was provided by Arla Foods Ingredients Group P/S (Lacprodan^®^ OPN-10; Arla Foods Ingredients Group P/S, Viby J, Denmark) with 99.5% purity of the protein component [21].

### 2.2. Bacterial Culture

*A. naeslundii* AK 6, *L. paracasei subsp. paracasei* DSM 20020 and *S. mitis* SK 24 were grown on blood agar plates (Statens Serum Institut, Copenhagen, Denmark) at 35 °C under aerobic conditions. Prior to adhesion experiments, organisms were transferred to Todd-Hewitt broth (THB, Roth, Karlsruhe, Germany) and cultivated at 35 °C until early stationary phase.

### 2.3. Bacterial Adhesion

Adhesion experiments were performed using 24-channel microfluidic flow cells with a polydimethylsiloxane surface (Bioflux EZ, fluxion Biosciences, San Franscisco, CA, USA). Stimulated saliva, processed as described by de Jong et al. [22], was diluted with PBS (1:2), titrated to pH 7 and flushed through the channels for 5 min with a flow rate of 100 µL/h (1 dyn/cm^2^). Thereafter, a salivary pellicle was allowed to form for 30 min under static conditions at 35 °C. Bacterial cultures were washed in fresh THB (pH 7; 4696 g, 5 min), adjusted to an optical density of 0.5 (550 nm) and mixed with three parts of salivary solution. Proteins were thawed at room temperature, pasteurized for 20 min at 80 °C in a water bath and added to the bacterial suspensions, yielding a final concentration of 50 µM. The molecular mass of the milk proteins was calculated from their amino acid sequences (GPMAW V.13.02; Lighthouse Data, Odense, Denmark). For total milk protein an average molecular proteins mass of 21 KDa was used (based on 80% casein and 20% whey proteins with β-lactoglobulin being the dominant whey protein component). PBS was used as the negative control. The bacterial suspensions were then injected into the channels and a flow rate of 10 µL/h (0.1 dyn/cm^2^) was applied for 1 h, at 35 °C. Thereafter, PBS was flushed through the channels for 30 min (100 µL/h; 1 dyn/cm^2^) to remove non-adherent cells. Experiments were carried out in biological triplicates.

### 2.4. Quantification of Bacterial Adhesion 

Adhering bacteria were imaged in the viewing chamber of the flow channels using a bright-field microscope (Zeiss Axio Vert A1, Jena, Germany) equipped with a 40× objective (EC Plan-NEOFLUAR, Zeiss, Jena, Germany). Per channel, nine images (1920 × 1440 pixels) were acquired in different fields of view. The images were cropped to 1440 × 960 pixels, manually cleared for artefacts and segmented in the software daime (V2.2.3) [23] based on intensity thresholding. In each image, the area covered by bacteria was calculated.

### 2.5. Statistical Analysis

The average area coverage was calculated for each bacterial strain and treatment and normalized to control treatment (PBS). The data were log-transformed and their normal distribution and homogeneity of variance were checked with Shapiro–Wilk and Levene’s tests, respectively. One-way analysis of variance (ANOVA) followed by Dunnett’s post hoc test was used to compare each of the treatments to the control group, for each of the bacterial strains. All statistical analysis were performed using the software R (V.4.4.1) [24] with the significance level (α) set at 0.05.

## 3. Results

All purified milk proteins were estimated to be >95% pure judged by Tris-tricine gel electrophoresis and/or reverse-phase chromatography. All three employed bacterial species adhered well to the flow cells when treated with the negative control PBS. Compared to PBS treatment, the adhesion of *L. paracasei subsp. paracasei* and *S. mitis* was significantly reduced by OPN, α_s1_-casein, β-casein, β-lactoglobulin and crude milk protein, with the most pronounced effects observed for OPN and β-casein. In contrast, the attachment of *A. naeslundii* was only affected by OPN and crude milk protein (Figure 1). κ-casein and α-lactalbumin did not significantly reduce the number of adhering cells for any of the tested organisms. The null hypothesis was thus partly rejected. Representative microscopy images are shown in Figure 2.

## 4. Discussion

This work aimed to compare the ability of OPN to prevent bacterial adhesion to saliva-coated surfaces with the principal milk proteins α_s1_-casein, β-casein, κ-casein, α-lactalbumin and β-lactoglobulin, as well as crude milk protein. Compared to most hitherto published studies on the anti-adhesive properties of milk proteins, we mimicked the conditions of bacterial attachment more closely, by providing a shear-controlled salivary flow with a rate that matches the one in the oral cavity [18]. Bacteria could thus not attach after passive sedimentation to the saliva-covered substrate but had to adhere actively against the medium flow. The molar concentration (50 µM) for all employed proteins was chosen based on previous experiments that had demonstrated a significant but not a maximum effect for 46 µM of OPN [4]. Other studies investigating the effect of milk proteins on bacterial attachment have operated with similar concentrations [11,14,25].

Our work confirmed the anti-adhesive effect of OPN, which was significant for all tested bacterial strains. Κ-casein, in contrast, did not reduce bacterial adhesion, which is in disagreement with some previously published reports [15,26] but in line with previous results of our group on casein glycomacropeptide (CGMP), the C-terminal fragment of κ-casein. The observed discrepancy may be explained by differences in the experimental setup. Both Vacca-Smith et al. and Schüpbach et al. tested bacterial adhesion under static conditions using higher concentrations of κ-casein (≥200 µM), which was added prior to bacterial inoculation and hence incorporated into the salivary pellicle.

In the present study, all other proteins reduced the amount of adhering *S. mitis* and *L. paracasei subsp. paracasei* to some extent, although the difference did not reach the level of statistical significance for α-lactalbumin. *Streptococcus* spp. and *Lactobacillus* spp. both bind salivary receptors via cell-wall attached serine-rich repeat proteins (SSRPs) [27], but the fact that inhibition was achieved by different milk proteins suggests an unspecific mode of action rather than a targeted binding to particular bacterial adhesins. OPN has previously been shown to confer hydrophilicity to bacterial surfaces [5], and hydrophilic bacteria have repeatedly been demonstrated to be less capable of attaching to saliva/dental tissues [28,29,30,31]. The same mechanism may explain the reduced bacterial attachment upon treatment with other milk proteins, most of which possess strongly hydrophilic domains [32]. Future work may investigate the binding of the major milk proteins to different oral biofilm formers and their effect on surface hydrophobicity. 

Contrary to *Lactobacillus* spp. and most Streptococci, *A. naeslundii* possesses type I fimbria, long (~700 nm) cell appendages that mediate binding to statherin and proline-rich proteins (PRPs) [33]. Type I fimbria are thus involved in binding to saliva-coated surfaces [34,35], which was illustrated by an unusually long rupture length in atomic force microscopy (AFM) retraction experiments [4]. Neither α_s1_-casein, nor α-lactalbumin or β-lactoglobulin had any noteworthy effect on the adhesion of *A. naeslundii*. β-casein reduced the number of attached cells but not significantly. The effect of OPN was significant, but a previous study had shown that *Actinomyces* spp. required higher concentrations to prevent adhesion than Streptococci [4]. It is conceivable that *A. naeslundii*’s fimbriae offer fewer binding sites for milk proteins and that they penetrate the hydrophilic barrier on the residual cell surface. The theory is supported by rodent experiments, which showed that treatment with caseinate or skim milk shifted the microbial composition in plaque towards *Actinomyces* [36].

Interestingly, the effect of treatment with skim milk protein almost matched the one of OPN for all strains in the present study. Since caseins and β-lactoglobulin constitute the bulk of the protein mixture [37] and OPN is only present in negligible amounts, these results suggest a synergistic action of several milk proteins on different bacterial adhesins. Future work may investigate the exact mechanisms by which milk proteins bind different bacteria and identify a product with an optimal composition to hamper bacterial attachment to dental tissues.

In summary, this work demonstrates the ability of different milk proteins to reduce bacterial adhesion to saliva-coated surfaces, with OPN being the most effective single protein. The administration of milk proteins as part of oral care is a promising ecologic approach that may delay the formation of dental biofilms and the onset of biofilm-related diseases without harmful side effects on the commensal oral microbiota.

## Figures and Tables

**Figure 1 biomedicines-10-01922-f001:**
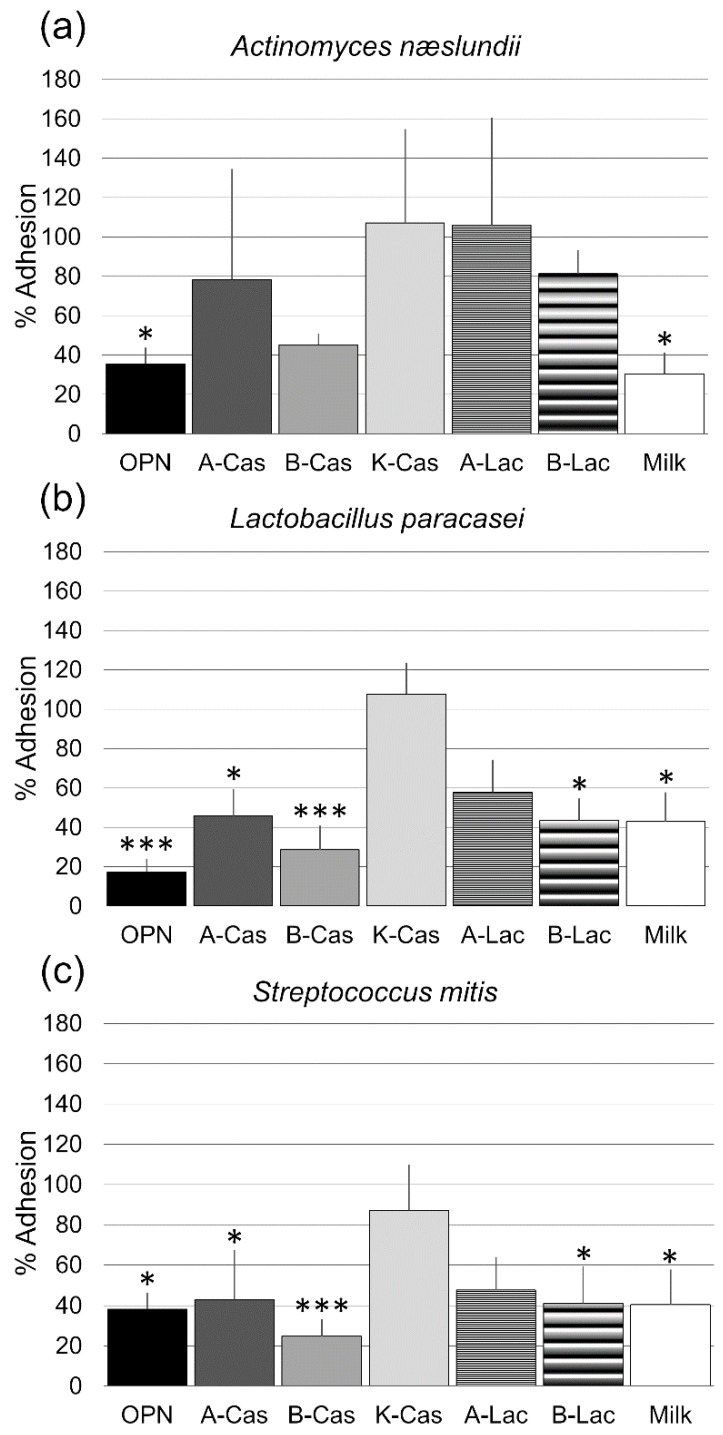
Effect of osteopontin (OPN), α_s1_-casein (A-Cas), β-casein (B-Cas), κ-casein (K-Cas), α-lactalbumin (A-Lac), β-lactoglobulin (B-Lac) and crude skim milk protein (Milk) on the adhesion of *Actinomyces naeslundii* (**a**), *Lactobacillus paracasei* (**b**) and *Streptococcus mitis* (**c**). Summary data from three biological replicates. For each treatment, adhesion was quantified in nine fields of view. Bars normalized to control group. Error bars = SD. * *p* < 0.05; ** *p* < 0.01, *** *p* < 0.001 (One-way ANOVA followed by Dunnett’s post hoc test).

**Figure 2 biomedicines-10-01922-f002:**
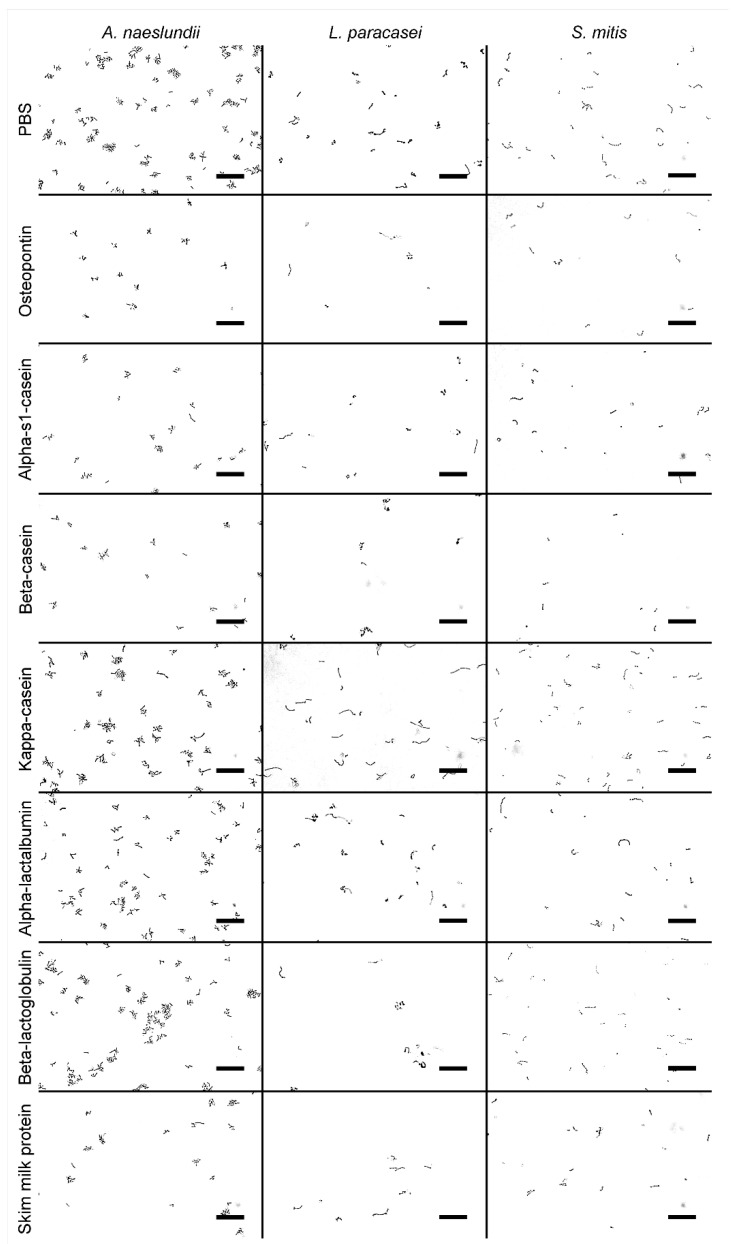
Representative bright field images of adherent bacterial cells. Only osteopontin and crude skim milk protein were able to significantly reduce the attachment of all three bacterial species. PBS = phosphate buffered saline. Bars = 20 µm.

## Data Availability

The data presented in this study are available on request from the corresponding author.

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
