# Peer review of "Prevention of Initial Bacterial Attachment by Osteopontin and Other Bioactive Milk Proteins"

_biomedicines, 2022, doi:10.3390/biomedicines10081922_

Round 1
Reviewer 1 Report
The authors discuss bioactive milk proteins that show the potential to delay harmful biofilm formation on teeth and hence the onset of biofilm-related oral disease, outlining the role of osteopontin. I consider a well-written manuscript that may be accepted in the current variant.
Author Response
Thank you for your time and attention.
Reviewer 2 Report
In this manuscript, Kristensen and colleagues demonstrated that Osteopontin, a matricellular protein expressed in most mammalian tissues and body fluids, prevent the adhesion of bacteria to saliva-coated surfaces under shear-controlled flow conditions. Their rationale is based on very recently observation where it has been proposed that bovine milk OPN can be used as therapeutic agent to prevent the generation of dental biofilms which is the responsible for the development of caries lesions. The milk proteins are one of the most promising resources used to hamper the bacteria adhesion to teeth without affecting the microbial homeostasis in the mouth. Moreover, they demonstrated that OPN alone was the most effective protein to reduce the adhesion of Actinomyces naeslundii, Lactobacillus paracasei subsp. paracasei and Streptococcus mitis.
It is a very interesting manuscript that can lay the bases for use OPN as a protein that can be administered as a part of oral care in order to reduce the formation of caries lesions.
This manuscript can be suitable for publications after minor revisions.
MINOR:
· In line 44, the authors used the world “microbiome” referring to the commensal bacteria present in the mouth. The world is not correct since “microbiome” refers to the collection of genomes of all the microorganisms found in a particular environment. The authors should replace this world using “microbiota”.
· In figure 1a, 1b and 1c, the description of the Y axis is not present. Please the authors should add this part
Author Response
We'd like to thank you for your time and consideration. All the requested changes have been included in the revised manuscript.